# Implementation of M-Chat for Screening of Early Signs of Autism in the Brazilian Health Care System: A Feasibility Study

**DOI:** 10.3390/nursrep15040120

**Published:** 2025-03-31

**Authors:** Nadia Maria Giaretta, Sabrina Pires Trufeli, Felipe Alckmin-Carvalho, Maria Cristina Triguero Veloz Teixeira

**Affiliations:** 1Graduate Program in Human Developmental Sciences, Mackenzie Presbyterian University, Rua da Consolação, 930, Consolação, São Paulo 01392-907, Brazil; nagiaretta@gmail.com (N.M.G.); 10339640@mackenzista.com.br (S.P.T.); mariacristina.teixeira@mackenzie.br (M.C.T.V.T.); 2Department of Psychology and Education, Faculty of Social and Human Sciences, University of Beira Interior, Polo IV, 6200-209 Covilhã, Portugal

**Keywords:** autism spectrum disorder, M-CHAT, primary health care, early detection, public health, nursing professionals

## Abstract

**Background**: Although screening for early signs of autism spectrum disorder (ASD) using the Modified Checklist for Autism in Toddlers (M-CHAT) has been recommended by the Brazilian Ministry of Health since 2014, the feasibility of this intervention and its effects in primary care have not been sufficiently investigated. **Objectives**: (1) to verify the feasibility of implementing the M-CHAT in the Brazilian Unified Health System, through routine childcare vaccination; (2) to evaluate the level of knowledge and practices of nursing professionals in Brazilian primary health care in relation to ASD to check the expertise to apply M-CHAT to verify early signs of this condition; (3) to verify, after implementation, the frequency of children with possible early signs of ASD using the M-CHAT. **Methods**: This is an observational, cross-sectional study conducted in a medium-sized city in southeastern Brazil. A total of 97 nursing professionals from 21 health facilities participated. The professionals answered a questionnaire on knowledge and practices regarding ASD, attended training on early signs of ASD and for the use of M-CHAT. Finally, they administered the M-CHAT to 267 parents of children aged 16 to 57 months in primary care settings. **Results**: Insufficient knowledge of ASD was identified among the nursing professionals evaluated. Approximately 80% agreed that the training was satisfactory, and 88% agreed that they felt competent to use the M-CHAT; 74% agreed with the benefits of using the M-CHAT to detect early signs of ASD in public health settings. However, 91% of the professionals reported that incorporating the M-CHAT into the day care routine overloaded their work, and less than 50% agreed that the instrument should continue to be used in the day care routine. Sixty-seven (25.09%) children screened positive for possible early signs of ASD by M-CHAT. **Conclusions**: The insufficient level of knowledge on ASD found among nursing professionals suggests gaps in Brazilian academic and professional training in this area. Although most participants recognized the importance of early detection of signs of ASD in primary care settings, the implementation of the M-CHAT in the routine primary care settings was not well accepted for most participants due to work overload, which indicates the need for public health policies to offer working conditions that minimize the overload of professionals, maintaining early screening interventions for ASD in Brazilian primary care institutions.

## 1. Introduction

Autism Spectrum Disorder (ASD) is classified under the category of Neurodevelopmental condition in the 5th revised edition of the Diagnostic and Statistical Manual of Mental Disorders (DSM-5-TR) [1]. The clinical features that support the diagnosis are persistent impairments in reciprocal social communication and social interaction as well as restricted and repetitive patterns of behaviors, interests, or activities [1]. Signs and symptoms must be present from early childhood and limit or impair daily functioning [1]. The symptoms associated with ASD are usually recognized during the second year of life, between 12 and 24 months; however, in severe cases, these signs can be observed before 12 months of age [1]. It is important to consider that a diagnosis of ASD in an individual with intellectual disability is appropriate when social communication and interaction are significantly impaired relative to the developmental level of the individual’s nonverbal skills [1]. However, intellectual disability could be an appropriate diagnosis when there is no apparent discrepancy between the level of social-communicative skills and other intellectual skills [1].

It is well established in the literature that early identification and intensive stimulation of children diagnosed with ASD is associated with better prognosis. The responsiveness of the nervous system to stimulation in the first two years of life produces more pronounced effects due to the maximum plasticity of the brain in the early stages of development [2,3,4,5]. Therefore, the care and stimulation of children in the first years of life allows greater gains in neuropsychomotor development, such as significant gains in intelligence coefficient and language development for communication purposes. All these improvements allow for better adaptive functioning and greater autonomy for the child, which has a positive impact on the quality of life of the individual with ASD, as well as their caregivers and family members [6].

Previous scientific evidence indicates that children with ASD already present signs of ASD between 12 and 24 months. A study of Backes et al. [7] revealed, from the analysis of 171 episodes of home videos made available by parents and filmed before the diagnosis of ASD, that the loss of words, deficits in social skills, and impairments in playing and socialization are indicators and warning signs for ASD. Alhozyel et al. [8] examined the association of commonly measured early developmental milestones with later diagnosis of ASD comparing 280 children with ASD and 560 typically developed children matched to cases by date of birth, sex, and ethnicity. 

The findings highlight the potential role of early developmental milestones as early signs of ASD that could facilitate earlier referral and diagnosis of ASD. It is essential that children between 12 and 24 months of age be monitored to identify developmental delays and signs of risk for ASD [9]. In developing countries, these actions in childcare routines are very important, considering that in these countries exposure to perinatal risk factors use to be higher compared to developed countries; for example, preterm birth, low birth weight, hyperbilirubinemia clustering of pregnancy complications, and maternal immigrant status [10].

Different technologies have been tested for the earlier diagnosis of ASD, including the use of innovative machine learning and deep learning technologies [11,12,13] especially in developed countries. However, in underdeveloped or developing countries, even screening for ASD with low-cost instruments that assess child development based on the reports of parents and caregivers of children at an early age remains a health challenge. In these countries, one of the main barriers to early diagnosis of ASD is the implementation of screening and monitoring actions in public health services for the mental health of children in early childhood, even when public policies that advocate such early screening already exist.

The diagnosis of ASD, especially in early stages and in poor countries, remains challenging. The main reasons for this difficulty include the heterogeneity of clinical features [14,15,16,17] and the lack of consensus on what constitutes a diagnosis of ASD [14,15,16,17]. For example, a large and recent epidemiologic study found that most cases of evaluated ASD (70%) do not present intellectual disability as a comorbidity, which is a different nosologic entity [18]. Especially in these cases, which are the majority, the absence of significant cognitive impairments may have less impact on the different contexts in which the child is inserted and may, therefore, be less obvious. This makes seeking help less likely and can make diagnosis more difficult. Another barrier to early identification of ASD is the difficulty of access to health services for the population [19,20,21], the lack of health professionals properly trained in recognizing indicators of child development and early signs of ASD [22], and the irregular and non-standardized implementation of routine to assess early signs of ASD [22,23].

Considering this scenario, it is important that, on a micro-regional scale, policy makers and practitioners test strategies that can provide early screening for ASD and verify whether these strategies can in fact be successfully implemented facilitating effective planning and future investments. 

The study of Araripe et al. [23] identified the profile of service use, barriers to access care, and sociodemographic factors related to these barriers in Brazilian families of children with ASD (sample of 927 families with children with ASD between 3 and 17 years from five Brazilian regions). The study verified a high variability of use of mental health services according to age, the region of residence, type of health care system used, and the parents/caregivers’ education, but the access to behavioral interventions was more frequent among users of the private system/health insurance and families whose caregivers had higher education [23]. Montiel-Nava et al. [24] verified the age of parents’ first concerns about their children with ASD and compared them with the age of ASD diagnosis (sample of 2520 caregivers of children with ASD in six different Latin America and Caribbean Countries). Results indicated that, on average, caregivers were concerned about their child’s development by 22 months of age; however, the diagnosis was received when the child was 46 months of age. In Brazil, with 1000 cases evaluated, the mean age at diagnosis was 47.3 months, well beyond the 36 months, the age at which diagnosis is considered early, suggesting that a window of opportunity for intensive stimulation has been missed in the country [22].

Still regarding the screening for early signs of ASD in Brazil, in a recent epidemiological study, Girianelli et al. [25] investigated the prevalence of early diagnosis of ASD and other types of pervasive developmental mental conditions in children receiving care at Psychosocial Care Centre for Children and Adolescents (CAPSi) (Psychosocial Care Centers (CAPS) are mental health care facilities of the “Sistema Único de Saúde (SUS)”, the Brazilian public health system whose main principles are universality, equity, regionalization, and decentralization. For more information on the Brazilian public health system in the area of mental health, see Onocko-Campos [26]. CAPS are health institutions intended for the diagnosis, treatment, and rehabilitation of people with severe mental health problems who cannot be cared for in the primary health care facilities [27]. The “CAPSi” model caters for children and adolescents who suffer intense psychological distress as a result of severe and persistent mental health problems that make it difficult or impossible to establish social ties and carry out life projects. This health service is indicated for municipalities or regions with a population of over 70,000 inhabitants [27]), in the five regions of Brazil. Onocko-Campos et al. evaluated data from the Record Outpatient Health Actions, between 2013 and 2019 [26]. Information on 22,483 children was analyzed. The authors found that early diagnosis of ASD and other developmental conditions (considered 48 months before in this study) occurred in 30% of cases during the period analyzed. They also found that the proportion of cases diagnosed early in Brazil is gradually increasing (from 23.3% in 2013 to 32.8% in 2019), with a statistically significant linear trend (*p* < 0.001), but it is still low. Analysis of the results also indicated that referrals from SUS primary care services, such as childcare, to CAPSi were associated with higher proportions of early diagnosed ASD cases, compared to spontaneous demand from caregivers and family members directly at CAPSi. This finding highlights the important role of primary care health professionals in the initial assessment of possible neurodevelopmental conditions and the referral of cases for specialized evaluation [26].

In Brazil, the document “Guidelines for Rehabilitation Care for People with ASD” presents a set of expected child development indicators for early age groups (from 0 to 48 months), as well as indicators of delays in neuropsychomotor development and early signs of ASD [28]. The document is a valuable tool to be used by primary care teams, bearing in mind that to identify a sign of the conditions, it is essential that the professional also recognizes the expected developmental milestone for the age in each area. The guideline aims to provide guidelines for multi-professional teams who provide care for children and adolescents in the various SUS health services [28]. The Brazilian Ministry of Health has presented a set of guidelines for primary care health service professionals to monitor the development of children in the first three years of life, with the aim of detecting early signs of ASD, identifying suspected cases, and referring them for diagnostic assessments by a specialized multidisciplinary team [28].

However, previous studies have already shown that primary health care professionals, including nursing professionals, psychologists, pediatricians, and social workers, need continuing training to be able to identify early signs of ASD in toddlers [19,29]. Discussions and actions regarding the need to train health professionals arise in the context of various problems, such as gaps in academic training, lack of government investment, and lack of preparation due to the scarcity of training and resources for the neuropsychomotor process of child development or the inappropriate use of neuropsychomotor development assessment tools [30]. Curricular structures of undergraduate courses in health areas have already revealed a lack of content on child development and signs of neurodevelopmental conditions and other psychiatric disorders [31,32,33]. Even courses whose basic object of study is mental health, such as psychology, are deficient in terms of training in this area [33,34]. 

Specifically, regarding nursing professionals’ knowledge of ASD, the focus of this study, a recent scoping review [35], which analyzed the results of 10 Brazilian studies, found that these professionals had insufficient knowledge of the neurodevelopmental issues typically associated with ASD. Participants reported few opportunities for training in this area, both during their academic training and in the health services where they worked. These gaps increase the risk of late diagnosis and intervention or even misdiagnosis [36], which could be avoided with adequate training. Furthermore, previous Brazilian studies show that both in primary care and in specialized outpatient services, nursing professionals, pediatricians, and other health professionals lack preparation to perform clinical assessments in suspected cases of various neurodevelopmental conditions, including ASD [22,24]. Although primary care is a privileged level of the SUS health care networks in Brazil, it is necessary to equip professionals to play this role and offer assessment models that can be included in routine actions to monitor the health of children at an early age. 

Backes et al. [37] conducted a systematic review of the psychometric properties of instruments used to assess ASD in the Brazilian population. The authors found six instruments validated to Brazilian population: Childhood Autism Rating Scale (CARS), the Autism Behavior Checklist (ABC), the Autism Screening Questionnaire (ASQ), the Autism Diagnostic Interview-Revised (ADI-R), the Autistic Traits Assessment Scale (ATA), and the Modified Checklist for Autism in Toddlers (M-CHAT). Analysis of the instruments available in Brazil revealed that when it comes to screening for early signs of ASD in a non-clinical population, the M-CHAT was the best adapted, with acceptable internal consistency scores. In a more recent systematic review of instruments for detecting early signs of ASD in non-clinical populations, Seize et al. [38] identified 11 instruments. However, only the M-CHAT had evidence of its validity demonstrated for the Brazilian population.

The M-CHAT is designed to screen early signs of ASD in children 16–24 months of age, based on reports from parents or caregivers. It is a simple, quick to administer instrument that can be administered by any trained health professional. Items are scored with “yes” or “no” answers and assess the following areas: social interaction, eye contact, imitative behavior, and ability to pretend and use eye contact and gestures to direct the partner’s social attention or to ask for help. The original instrument showed good indicators of internal validity (α = 0.85), with sensitivity of 0.97 and specificity of 0.95 [39]. The cross-cultural adaptation study of the M-CHAT for the Brazilian population was conducted by Losapio and Pondé [40], and the study evaluating the psychometric properties of the instrument was conducted by Castro-Souza [41], who found a Cronbach alpha of 0.95, a sensitivity of 0.94 and a specificity of 0.91.

The screening for early signs of ASD using the M-CHAT has been recommended by the Brazilian Ministry of Health since 2014 [28,42]. However, the feasibility of this measure has been little investigated in Brazilian public health services. Given the epidemiological relevance of ASD and the importance of early diagnosis and intervention for a better prognosis, the main objective of this study was to verify the feasibility of implementing the M-CHAT in the Brazilian Unified Health System, through routine childcare vaccination. Additionally, we evaluated the level of knowledge and practices of nursing professionals in Brazilian primary health care in relation to ASD to check the expertise to apply M-CHAT to verify early signs of this conditions, and verified, after implementation, the frequency of children with possible early signs of ASD using the M-CHAT.

## 2. Methods

### 2.1. Study Design and Setting

The study design was observational and cross-sectional. The study was carried out in the municipality of Cotia, São Paulo state and located in the southeastern region of Brazil, which is the most developed in the country in terms of income and human development. According to Brazilian Institute of Geography and Statistics [43], the population of Cotia in 2021 was 257,882 inhabitants. In the last assessment, conducted in 2010, Cotia had a Human Development Index of 0.780, considered high compared to other Brazilian municipalities; however, with high levels of social inequality [44]. Although the gross domestic product (GDP) of the city of Cotia was R$62,486.58 (EUR: 10,243.60) in 2021 [43], much higher than Brazil’s GDP per capita in the same year, which was R$42,247.52 (EUR: 6925.82), income indicators alone do not guarantee human development, since Brazil is one of the countries with the greatest social inequality assessed by the GINI Index [45] in the world, and even in the most developed states, such as São Paulo, there are families in situations of socioeconomic vulnerability and food insecurity, especially on the outskirts of medium and large urban centers.

### 2.2. Participants

Two groups took part in the study: (1) nursing professionals (auxiliary, technician, registered nurse) and (2) parents or caregivers of children (non-clinical sample) attended at 21 of the 26 Basic Health units of SUS, in the city of Cotia, in 2021. This was a non-probabilistic sample selected according to convenience criteria. The main characteristics of the two groups that took part in the study are described below.
(1)Nursing professionals: this group consisted of 97 (91.5%) nursing professionals out of a total of 106 eligible professionals working in public primary health care institutions in Cotia. The nursing professionals were nursing auxiliary (n = 12), technicians (n = 46), and registered nurses (n = 39). Most of participants were female (87.23%), and mean age was 44.54 years old (SD = 9.32). Most professionals had the highest level of technical education (n = 47; 48.45%), followed by higher education (n = 27; 27.84%) and higher education and lato sensu postgraduate studies (n = 23; 23.71%). More than half of the nursing professionals (n = 56; 59.57%) had worked in the health sector for more than five years, followed by those who had worked for between one and three years (n = 20; 28%), less than one year (n = 12; 12.77%), and three to five years (n = 6; 6.38%). Although the nursing professionals work at different levels (auxiliary, technician, or nurse), they performed the same functions in the basic health units. They all carried out nursing procedures, such as dressings, administering medication, vaccinations, collecting material for tests, washing, and preparing and sterilizing materials.(2)Parents or caregivers: between November 2019 and January 2021, 267 mothers or main caregivers, from an eligible sample of 3601 children, attended childcare and vaccination sessions at the 21 health units. Thus, the sample for this study consisted of 267 primary caregivers. The inclusion criterion for mothers or caregivers was that they cared for their child for at least 6 h a day. The mean age of children was 22.39 years old (SD = 4.23), and 50.94% were male.

### 2.3. Instruments

(a)Questionnaire to assess knowledge of typical developmental milestones and early warning signs for ASD: developed by the researchers for this study, the instrument assesses expected developmental milestones and early warning signs for ASD in early childhood, predominantly between 12 and 30 months. It was based on the Ministry of Health’s document “Guidelines for the Rehabilitation of People with Autism Spectrum Disorders” [28]. The questionnaire is structured in two parts: the first, made up of 14 multiple-choice questions, assesses professionals’ knowledge of child development; the second, made up of 8 questions, assesses the conduct expected in primary care health services in cases where possible developmental alterations are identified. The validity of the instrument was checked by three independent judges, all of whom had a doctorate in human development, as well as clinical and teaching experience in developmental disorders. The content validity coefficient technique was used to assess the clarity, precision, and objectivity of the instrument [46]. The index of agreement between judges was calculated using the kappa coefficient and showed agreement of 0.86 in objectivity and 0.87 in precision criteria, considered satisfactory according to Pasquali [47]. However, the clarity criterion score was 0.66, considered relatively low. Proposed changes were made to the items to create an instrument whose items had adequate criteria for precision, clarity, and objectivity, considering the type of construct validity adopted.(b)The “Modified Checklist for Autism in Toddlers” (M-CHAT): developed by Robins et al. [39], the instrument comprises 23 items designed to screen for early signs of ASD in children aged between 16 and 24 months, based on reports from parents or caregivers. Any previously trained health professional can use it. The items are scored with “yes” or “no” answers to indicate the presence of signs of ASD. The items assess the child’s interest in social interaction, their ability to maintain eye contact, their aptitude for imitation, their tendency to play make-believe and to use eye contact and gestures to direct their partner’s social attention or to ask for help. The M-CHAT scoring algorithm states that, except for items 2, 5, and 12, a “no” answer indicates a risk of ASD, while a “yes” answer indicates a risk of ASD for items 2, 5 and 12. According to the instrument’s correction indication, a total score of 4 or more, or 2 or more positive critical items, indicates a risk of ASD, which suggests the need for more in-depth assessments by specialized professionals. In the original study evaluating the psychometric properties of the M-CHAT, internal reliability was found to be adequate for both the entire checklist and for the critical items (α = 0.85 and α = 0.83, respectively). Sensitivity was 0.97, and specificity was 0.95, both considered high [39]. The cross-cultural adaptation study of the M-CHAT for the Brazilian population was carried out by Losapio and Pondé [40], and the study evaluating the psychometric properties of the instrument for the Brazilian population was carried out by Castro-Souza [41], who found an alpha of 0.95, a sensitivity of 0.94, and a specificity of 0.91.(c)Questionnaire to assess the training of professionals in the use of the M-CHAT in-strument and evaluation of its implementation in vaccination childcare routines, con-sisting of seven questions that assessed, according to the nursing professionals’ reports, their perceptions of the quality of training and support actions for the use of the M-CHAT in childcare and immunization routines, the perception of overload, and the evaluation of the maintenance of the M-CHAT as part of the routines.

### 2.4. Training for Nursing Professionals

Training consisted of 4 h theoretical course on typical developmental milestones and early warning signs of ASD. The presentation, in power point format, was developed by the first author of the study, as part of a doctorate in Developmental Disorders, and supervised by the last author, who holds a PhD in Health Sciences. The content of the slides was based on the document “Guidelines for the Rehabilitation of People with Autism Spectrum Disorders” [28] and on DSM-5 [1]. The slides were accompanied by videos illustrating expected developmental milestones and warning signs of ASD in each of the clinical domains of the DSM-5 diagnostic guidelines [1]. The topics covered were early signs of this condition, screening instruments, epidemiology, and main evidence-based interventions for the care of children with ASD.

### 2.5. Proceedings

The first author of the study mapped and contacted the coordinators and/or directors of the 26 primary health care units in Cotia, of which 21 (80.7%) agreed to take part in the study. The participating primary health care professionals were nominated by the local coordinators, considering their availability to take part in the study, so that there was no impact on the work routine in the institutions. The questionnaire assessing knowledge and practices on ASD was completed individually, during the nursing professionals’ work breaks, and lasted approximately 15 min. The training on expected developmental milestones and warning signs of ASD in early childhood was carried out in the institutions, in groups of 5 to 15 nursing professionals, during working hours and lasted 4 h. The M-CHAT was administered to the children’s parents or caregivers after the vaccination appointment, and lasted approximately 10 min.

### 2.6. Data Analysis

Descriptive analyses were carried out on the simple frequency and proportion of correct answers given by the nursing professionals to the items related to knowledge and practice questions about ASD. Analysis of variance (ANOVA), Tukey’s post hoc test and Cohen’s d-test were used to assess the effect size of differences. The chi-squared test was applied to assess possible age and gender differences between children assessed as being at high and low risk of ASD.

### 2.7. Ethical Aspects

This study was approved by the Research Ethics Committee of the Mackenzie Presbyterian University, São Paulo, Brazil (number: 867.042, CAAE: 36991614.0.0000.0084; 11 November 2014). All the participants provided written informed consent. The study was conducted adhering to the ethical principles established by the Declaration of Helsinki.

## 3. Results

In the sample of nursing professionals assessed on their knowledge of ASD, which could range from 0 to 14 points, we found an average of 5.24 (SD = 2.46). No professional answered all the questions correctly, and the maximum score was 11 points. We found that no question was answered correctly by more than 70% of the nursing professionals assessed. Table 1 shows the frequency and percentage of nursing professionals who answered each question of the ASD knowledge questionnaire correctly.

The questions with the highest frequency of correct answers referred to signs and symptoms of ASD, symptoms of ASD that are sensitive to medication and the types of therapies recommended for the care of children diagnosed with ASD, and the composition of the multidisciplinary team for the clinical assessment of the diagnosis. Less than 20% of the professionals assessed correctly answered the questions related to the epidemiology of ASD, ASD comorbidities with genetic syndromes, and ASD symptoms. Approximately one in four professionals knew about tools for screening for early signs of ASD. Table 2 shows the frequency and proportion of nursing professionals who answered each question of the ASD practice questionnaire correctly.

Regarding the scores for practices related to ASD, which could vary between 0 and 8 points, we found an average of 5.64 (SD = 1.95). The questions with the highest frequency of correct answers (90%) referred to guidance for parents when the nursing professional notices any language alterations or changes in play behavior, and guidance for parents to seek specialized care when they notice behaviors that may indicate some neurodevelopmental alteration. The lowest scores were found for question 1, which assessed the professional’s self-perception of their skills and competencies in advising a parent or caregiver on the stages of neuropsychomotor development. Table 3 shows the results of the differences in the means of the knowledge and practice scores in relation to the type of training of the professionals (auxiliary, technician, or registered nurse).

The knowledge mean score was higher for registered nurses, which differed significantly from those trained as auxiliary or technician. The practice score revealed no significant difference when comparing the types of training, although, in descriptive terms, registered nurses also showed higher scores in this category. 

According to the M-CHAT, of the 267 children evaluated, 67 (25.09%) tested positive for possible early signs of ASD. The total M-CHAT scores of the 267 children ranged from 0 to 15 points, with a mean of 1.93 points (SD = 2.08). Regarding critical items, 209 children (78.28%) scored none, 44 children (16.48%) scored one critical item, eight children (3%) scored two critical items, three children (1.12%) scored three critical items, two children (0.75%) scored four critical items, and only one child (0.37%) scored five critical items. Although the proportion of boys with possible early signs of ASD was higher, there were no significant differences in relation to the proportion of girls. No significant differences were found between the M-CHAT score and the sex and age group of the children assessed. Table 4 shows the associations between the M-CHAT score and the sex and age group of the participants. 

Regarding the feasibility of including the M-CHAT in the childcare routines of health units, approximately 80% reported that the course fulfilled its objective of training them in knowledge about early signs of ASD, and 88% reported feeling able to use the M-CHAT. In addition, 74% agreed with the benefits of using the M-CHAT to detect early signs of ASD in public health. However, 91% of the professionals reported that incorporating the M-CHAT into the childcare routine overloaded their work, and less than 50% agreed that the instrument should continue to be used in childcare routines.

## 4. Discussion

Our aim in this study was to assess the level of knowledge and practices regarding ASD among nursing professionals working in primary health care in the Brazilian public health system, in a medium-sized city located in the state of São Paulo, in the southeastern region of Brazil. In addition, we evaluated the feasibility of incorporating the M-CHAT into the vaccination childcare routines, after training nursing professionals. Additionally, we evaluated the frequency of children with possible early signs of ASD using the M-CHAT. It is well established in the literature that screening for signs of ASD among children in the first years of life, associated with proper diagnosis and care based on the best available scientific evidence, when carried out early, is associated with better prognoses [4,6].

The M-CHAT has been described as the instrument with the best validity indicators [40,41] for screening early signs of ASD in children available for the Brazilian population [37,38]. Although it has been used in several Latin American countries, such as Uruguay, Argentina, and Chile, over the last decade [21,48,49], and its use has been recommended by the Brazilian Ministry of Health since 2014 [28,42], its use for screening for early signs of ASD in children is not without limitations. Pandey et al. [50] state that the M-CHAT appears to be more suitable for detecting signs of ASD from 24 months of age [50] and that the low specificity of the M-CHAT should be considered when screening children between 16 and 23 months of age. A systematic review with meta-analysis assessed the accuracy of the M-CHAT in 13 studies analyzing different versions of the instrument in different countries [51]. The authors found similar results to Pandey et al. [50], with high sensitivity (0.83) but relatively low specificity (0.51), especially when assessing children under two years of age.

With the large-scale application of the M-CHAT, due to its low specificity, the outcome could be many cases of false-positive results for ASD in evaluations carried out in basic health units. The referral of a very large number of cases for clinical assessment in more complex SUS services, such as CAPsi. This measure would overburden the Brazilian public health system, which is already historically underfunded and overloaded, especially regarding CAPSi, which are insufficient in number and unevenly distributed in the different regions of Brazil [52]. On the other hand, it could have a positive effect by identifying other neurodevelopmental delays that may share characteristics with ASD, or even children with typical development in situations of social vulnerability, whose neurodevelopmental alterations are due to inadequate contextual factors, such as food insecurity, affective and care neglect, and lack of environmental stimulation. 

Furthermore, it is worth mentioning that since 2014, the same year that the BrazilianMinistry of Health guidelines were published recommending the use of the M-CHAT to screen for signs of ASD in children in Brazil [28], there has been an updated version of the instrument, called the Modified Checklist for Autism in Toddlers, Revised with Follow-up (M-CHAT-R/F) [53]. The main difference between this instrument and its original version, apart from minor changes in content to make it more intelligible and the removal of some items to increase internal validity, is the inclusion of a follow-up interview for positive cases aimed at parents or caregivers, in which the items in which the child failed are investigated in greater depth. The aim of including the follow-up interview was to increase the specificity of the M-CHAT and, therefore, reduce the number of false-positive cases. In a recent systematic review with meta-analysis [54], in which 50 studies were evaluated, the authors found that the pooled sensitivity of M-CHAT(-R/F) was 0.83 (95% CI, 0.77–0.88), and the pooled specificity was 0.94 (95% CI, 0.89–0.97), which indicates an improvement in the psychometric properties of the instrument.

### 4.1. Feasibility and Evaluation of Training on ASD and M-CHAT

The training on neurodevelopmental conditions, identification of early signs of ASD, and on the application of the M-CHAT was well accepted and well evaluated by the nursing professionals. Although most professionals recognized the importance of detecting early signs of ASD in primary care settings, the introduction of the M-CHAT into the routine of the community basic health units was not well accepted. The burden associated with the inclusion of this intervention was the main reason cited by professionals for low acceptance. 

We found that approximately 40% of the nursing professionals who took part in the study had less than three years in the health sector. Historically in Brazil, nursing professionals have received low salaries in the public health system, especially in primary health care setting and in early stages of career [55,56,57]. According to law 2564/2020 [58], the national salary floor for nursing professionals, for a 40 h working week, should be R$2375.00 (EUR 389.34) (equivalent salary in Portugal, evaluated by the purchasing power parity calculator: EUR 522.03 for nursing auxiliary, EUR: 730.84 for technicians and EUR: 1044.06 for registered nurses) for nursing auxiliary, R$3325.00 (EUR 545.08) for technicians and R$4750.00 (EUR 778.68) for graduate nurses. Although the state of São Paulo, where the research was carried out, complies with the salary floor law for these professionals, the cost of living in this state is known to be higher, which can be a complicating factor in the household budget for these professionals. To get around these difficulties, a considerable number of these professionals must work double shifts, with excessive workloads that can exceed 60 h a week [56,59].

In addition, an accumulation of administrative and care duties [60], an excessive number of appointments and procedures, inadequate infrastructure in health units and a shortage of working materials are other drawbacks. These results indicate the need for public health policy leaders to provide continuity to this assessment model and to identify strategies and interventions that minimize the burden on professionals, with the aim of making it feasible to maintain early screening interventions for ASD in Brazilian primary care institutions.

Our results provide preliminary evidence that to include the M-CHAT in the childcare routines of primary health care institutions, some structural aspects need to be considered. The workload of nursing professionals could be reduced by hiring more professionals and reducing the administrative tasks that fall to this professional category in Brazil, diverting the focus away from health care work. In addition, relocating professionals from units with fewer users to those with greater demand would also help to relieve the overload and make it possible to include the instrument. The payment of fair salaries, appropriate to the level of responsibility and importance of the work of the nursing teams, could reduce the proportion of professionals working double shifts, reducing stress and increasing the availability of these professionals. All the measures presented as alternative solutions to the overload and the possibility of including M-CHAT in primary care routines imply an increase in federal budgetary investment in primary health care in Brazil.

Since the M-CHAT is a simple and quick-to-apply instrument, we believe that after training the entire nursing team, this measure could be carried out by auxiliaries, technicians and registered nurses. We assessed the feasibility of implementing M-CHAT considering the lower impact on care activities that are common to all the nursing professionals involved in this study. Although the number of technicians and auxiliaries was higher than that of registered nurses, most of the sample cited this work overload. Nursing professionals could be more affected by this overload, considering that, in addition to the duties that are similar to all of them, they can also carry out management activities.

### 4.2. Knowledge About ASD

Our main results indicated a low level of knowledge and little expertise in handling situations in which neurodevelopmental alterations are identified in children aged between 12 and 30 months by nursing professionals, especially among technicians and auxiliaries. Important knowledge for primary care professionals, such as the etiology of ASD, risk factors, prevalence, clinical indicators for differential diagnosis, and knowledge of ASD screening tools, showed low percentages of correct answers (up to 58% of correct answers). This result corroborates previous studies conducted in Brazil [61,62].

The practical questions that indirectly assessed knowledge of ASD (questions 1 to 4) showed that professionals perceived gaps in their academic training in relation to expected developmental milestones and recognizing early signs of ASD, as well as uncertainty in advising mothers and caregivers on what to do. This is consistent with the results of the first part of the questionnaire on general knowledge of ASD. The questions with the highest frequency of correct answers asked these professionals about their decision making in identifying language delays, impairments in play behavior and eye contact deficits. The answers to the questions about practices related to ASD were dichotomized as ‘yes’ and ‘no’, and the questions on the knowledge questionnaire were multiple choice, some with more than six possible answers. 

Our hypothesis is that the slightly higher scores for practice compared to knowledge of ASD may be related to the greater complexity of the multiple-choice questions in the ASD knowledge questionnaire compared to the dichotomized questions in the practice questionnaire. It may also be related to responses based on caution, i.e., when faced with signs or symptoms of neurodevelopmental abnormality, even with a high likelihood of a false positive, professionals tended to feel that the most appropriate course of action would be to refer the child for specialist assessment. This type of routine can contribute to early identification; however, false positive cases can overload the Brazilian public health system and increase the difficulty in absorbing the high flow of care.

### 4.3. Early Signs of ASD According to M-CHAT

We identified 25% of children with possible early signs of ASD. This result is similar to that found in underdeveloped or developing nations such as Indonesia, where Windiani et al. [63] found 24.54% of children with possible early signs of ASD, and much higher than what has been reported in studies carried out in developed countries. For example, in the American study by Kleinman et al. [64], the proportion of children identified with possible early signs of ASD using the M-CHAT was 11.63%, and in Spain, 17.29% [65]. We highlight that our result regarding a high proportion of cases with possible early signs of ASD should be analyzed with caution, since studies indicate that the high rates of false-positive cases may be associated with socioeconomic vulnerability, food insecurity or chronic nutritional deficiencies, as well to the lack of stimulation of children at early ages, adverse conditions that have gradually improved over the last few decades in Brazil, but which are still very prevalent in the country [66,67]. These adverse environmental variables limit adequate development even among children without ASD [68] and may also increase the number of false positives assessed by the M-CHAT in our study. In addition, false positives may be related to the lower educational level of parents or carers, or parents’ inaccurate report or they are not being willing or psychologically ready to endorse an increased likelihood of ASD behaviors during screening [54,69].

Moreover, researchers recommend that health professionals adopt a cautious approach to monitoring for early signs of ASD in very young toddlers, since some children with ASD, symptoms are subtle early in development or show a prolonged course of symptom development, consistent with the theory that although brain development may be different before birth, measurable symptoms of ASD may emerge gradually because children are expected, and fail, in demonstrating more sophisticated behavior as they grow older [16,17,54]. 

To minimize the possibility of identifying false-positives cases, multiple assessments of children’s neurodevelopment over time are recommended, using more than one screening tool and instruments that assess not only parents and caregivers reports on their children’s development, but also direct observation of behavior in structured situations by trained professionals [54].

Considering that M-CHAT has limitations regarding its reliability and potential for classification errors, in our study, the follow-up of cases identified with possible early signs of ASD by the M-CHAT was planned to be carried out by a multi-professional team specializing in neurodevelopmental conditions to refute or confirm the diagnosis of ASD. However, this planning was made impossible by the resurgence of the COVID-19 pandemic and social isolation measures.

### 4.4. Limitations and Future Directions

Although we achieved our objectives, our study has limitations that need to be presented. Our results represent an overview of Cotia, regarding knowledge and practices about ASD by nursing professionals, and the use of the M-CHAT to screen for early signs of ASD in primary health care in the Brazilian public health system. However, they may not reliably represent the condition of professionals in other regions of the state of São Paulo or other Brazilian states, given the cultural and economic heterogeneity of the country, which has continental dimensions. For this reason, we recommend carrying out multicentric studies in different Brazilian states to produce more robust evidence with greater external validity in terms of generalization.

In addition, the cross-sectional design of our study allowed us to present a momentary overview of the variables analyzed. As the research was carried out during the first months of the COVID-19 pandemic in Brazil, a period of great stress and uncertainty, our results on the acceptance of the implementation of M-CHAT in childcare routines may have been influenced by this variable. Due to the worsening of the COVID-19 pandemic in Brazil and social distancing measures, it was not possible to objectively reassess the level of knowledge and practices about ASD among health professionals after the training. Therefore, our assessment of the effects of the training was based solely on the professionals’ perception of the quality of the training, a result that may suffer from social desirability bias. Furthermore, the follow-up and reassessment of the cases identified by the M-CHAT by a specialized multidisciplinary team was also made impossible by the worsening of the COVID-19 pandemic and, therefore, it was not possible to confirm the diagnosis of ASD among the children assessed. 

For future studies, we suggest evaluating children at an earlier age and following up cases for diagnostic confirmation, to produce more evidence on the sensitivity and specificity of the M-CHAT for identifying ASD in Brazilian children. Cross-cultural adaptation studies and evaluation of the psychometric properties of the M-CHAT-R/F are needed. Subsequently, comparative longitudinal studies of the advantages and disadvantages of using the M-CHAT and M-CHAT-R/F, especially in terms of specificity and sensitivity, are recommended.

## 5. Conclusions

Nursing professionals of our sample showed insufficient knowledge regarding ASD, in terms of etiology, most prevalent signs and symptoms, epidemiology, screening tools, and care modalities, as well as little proficiency in guiding parents and caregivers of children with possible early signs of ASD. This result suggests the need to assess gaps in academic nursing training related to ASD, and the importance of continuing education on neurodevelopmental conditions. 

The training on neurodevelopmental conditions, identification of early signs of ASD and on the application of the M-CHAT was well accepted and well evaluated by the nursing professionals. We identified a high frequency of children with possible early signs of ASD as assessed by the M-CHAT. 

Although most professionals recognized the importance of detecting early signs of ASD in primary care settings, the introduction of the M-CHAT into the routine of the community basic health units was not completely accepted for all. The burden associated with the inclusion of this intervention was the main reason cited by professionals for low acceptance. These results indicate the need for public health policy leaders to provide continuity to this assessment model and to identify strategies and interventions that minimize the burden on professionals, with the aim of making it feasible to maintain early screening interventions for ASD in Brazilian primary care institutions. Childcare in primary care is a privileged place for detecting early signs of ASD. This study presents preliminary data on how to implement these actions through nursing professionals; a low-cost action that can be expanded and tested in Brazilian public health.

## Figures and Tables

**Table 1 nursrep-15-00120-t001:** Frequency and proportion of nursing professionals who correctly answered the questions on the ASD knowledge questionnaire (n = 97).

Question	Correct Answers	(%)
(1) What are the established etiological factors of ASD?	57	58.16%
(2) What are the signs and symptoms of ASD?	62	63.27%
(3) Which syndromes are usually associated with ASD?	14	14.29%
(4) Which symptoms of ASD are sensitive to drug interventions?	58	59.18%
(5) What therapies are recommended for interventions for children with ASD?	57	58.16%
(6) Identify which characteristics of a clinical case presented may be indicators of ASD	17	17.35%
(7) What is a savant skill?	21	21.43%
(8) Which areas are predominantly affected in people with ASD?	32	32.65%
(9) What is the approximate distribution of ASD cases by gender?	10	10.20%
(10) What is the best estimate of the prevalence of ASD in the general population?	12	12.24%
(11) Which of the ASD screening scales are you familiar with?	26	26.53%
(12) What is the main care unit in the public health network responsible for monitoring children with ASD?	51	52.04%
(13) What is the best composition of a multidisciplinary team for the clinical diagnostic evaluation of ASD?	58	59.18%
(14) What assessment procedures should be carried out when receiving a child with suspected ASD?	38	38.78%

**Table 2 nursrep-15-00120-t002:** Frequency and proportion of professionals who correctly answered the questions on the ASD practices questionnaire (n = 97).

Question	Correct Answers	(%)
(1) Feel safe in explaining the stages of neuropsychomotor development to a baby‘s parents?	37	37.76%
(2) Feel confident in assessing and communicating with the family about the stages of a baby‘s development?	55	56.12%
(3) Feel confident in alerting the family when the child doesn‘t speak at 28 months?	61	62.24%
(4) Feel confident in telling parents that their baby may have an indicator of altered development?	50	51.02%
(5) Do you guide a child‘s parents to a referral when you notice a language delay or change?	83	84.69%
(6) Do you ask for help from other professionals when the child doesn‘t know how to play as expected for their age with other children?	89	90.82%
(7) Do you recommend that parents see a specialist when they report that their child is acting differently?	89	90.82%
(8) Do you advise parents to seek specialized care when they notice that their child has difficulties and isn‘t saying anything?	89	90.82%

**Table 3 nursrep-15-00120-t003:** Comparison of mean knowledge and practice scores in relation to type of training (n = 97).

Groups	*M*	*DP*	*H*(*gl*)	*p*	Comparison	*p_Tukey_*	*d*
Knowledge							
Auxiliary (n = 12)	3.42	2.19	9.20(2)	0.01	Technical	0.09	0.72
Technician (n = 46)	5.02	2.25			Registered Nurse	0.17	0.39
Registered Nurse (n = 39)	5.95	2.47			Auxiliary	0.02	1.05
Practices							
Auxiliary (n = 12)	5.17	2.25	25.90(2)	0.31	Technical	0.92	0.12
Technician (n = 46)	5.41	2.02			Registered Nurse	0.35	0.31
Registered Nurse (n = 39)	6.00	1.75			Auxiliary	0.40	0.45

**Table 4 nursrep-15-00120-t004:** Association between M-CHAT score and gender and age group (n = 267).

**M-CHAT Classification**	**Sex of the Child**		** *X* ^2^ **	** *p* **
		**Male**		
No risk (n = 200)	101 (50.50%)	99 (49.50%)	0.45	0.50
At risk (n = 67)	30 (44.77%)	37 (55.23%)		
	**Child‘s Age Group**		** *X* ^2^ **	** *p* **
**M-CHAT Classification**	**16 to 22**	**23 to 30**		
No risk (n = 200)	105 (52.50%)	95 (47.50%)	0.45	0.50
At risk (n = 67)	39 (58.20%)	28 (31.80%)		

## Data Availability

Data will be available upon request.

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
