# Peer review of "Implementation of M-Chat for Screening of Early Signs of Autism in the Brazilian Health Care System: A Feasibility Study"

_nursrep, 2025, doi:10.3390/nursrep15040120_

Round 1
Reviewer 1 Report
Comments and Suggestions for Authors
Dear authors,
Thank you for the opportunity to read and learn with your manuscript. It is original and brings contributions to improving ASD diagnosis. However, some points need attention:
i) for me, the main aim of the article should be to verify the feasibility of implementing M-chat in the Brazilian Health System. The other objectives should be used to write another article. Thus, I would explore more the Brazilian health laws to give the reader the knowledge of "HOW" it could be done.
ii) the introduction is confused and barely describes the Brazilian scenario, which is the core of the manuscript. It would be interesting to describe the laws and history of how M-Chat was suggested to be a screening tool in Brazil.
iiI) the methodology needs to be improved. The information about the groups that are part of the study are too far from each other. I had to read the manuscript 4 times to understand that nursing professionals are one group and parents are the other. Also, there are some errors that need to be checked. At line 190 is written "At least two professionals from each of the 26 health units in Cotia took 190 part in the study", however, in line 177-179 there is different information "Our study was conducted with 177 nursing professionals and parents or caregivers of toddlers from 21 of the 26 Basic Health 178 units of Brazilian public health system in Cotia in 2021" Also about methodology, the sample was composed of 3 types of nursing professionals (auxiliary, technician and nurse). Although they work in the same area, it is not possible to analyze them as a group "nursing", as they have different functions, knowledge, work experience, that is, they are different groups. Regarding the instrument, the questionnaire was validated by professionals, but the statistical test used was not mentioned. It should be mentioned.
My suggestion to the writing is to reduce the objectives and focus on the feasibility of implementing M-chat in the Brazilian Health System and discuss who, which professional (auxiliary, technician or nurse) or other, would have the expertise to apply M-chat to verify early signs of ASD.
Author Response
Dear Reviewer 1, please find below the document with point-by-point response to your comments.
Best wishes,
Felipe Alckmin-Carvalho

Reviewer 2 Report
Comments and Suggestions for Authors
Dear Authors,
I would like to start by thanking you for your work. Below, I outline a few observations and suggestions that I believe could enhance the clarity, rigor, and impact of your manuscript.
- In line 41, please note that the correct abbreviation is DSM-5-TR as this refers to the revised version.
- The language used in relation to autism spectrum disorder (ASD) is somewhat outdated. For instance, in line 50, the term treating is used. It is important to acknowledge that ASD is a neurodevelopmental condition, not a disorder that is treated or cured. A revision of such terminology throughout the manuscript would align the study with contemporary neurodiversity research.
- The introduction would benefit from a discussion on the differential diagnosis between Intellectual Disability (ID) and ASD. The manuscript currently implies an implicit association between the two, which reflects outdated perspectives from the 1980s. Clarifying this would strengthen the theoretical foundation of the study.
- While the paper correctly highlights the lack of consensus regarding ASD identification, the M-CHAT is presented as a particularly effective tool without sufficient critical analysis. It would be valuable to introduce and compare other screening tools to contextualize the advantages and limitations of M-CHAT.
- Providing statistical insights regarding M-CHAT’s ability to discriminate ASD from other neurodevelopmental conditions (e.g., false positives, limitations in detecting different ASD profiles) would enhance transparency. A brief historical overview of the M-CHAT, its development, its known limitations, and the justification for studying it despite governmental recommendations would provide essential context. For example, it is useful to discuss how other countries (such as France) have historically recommended controversial approaches (ABA, packing) until the last few years, highlighting the need for a critical assessment of governmental guidelines.
- The manuscript repeatedly states that nurses already have excessive workloads. However, the study explores the possibility of adding new responsibilities to their tasks. Would this be sustainable? Could the study discuss alternative solutions, such as redistributing tasks or removing less essential duties, to prevent overburdening these professionals?
- More details on the training and disciplinary backgrounds of nurses in Brazil would provide better insights into their preparedness for autism screening and how feasible the proposed screening approach is in the real-world healthcare setting.
- The paper provides income data converted into euros. However, for a more meaningful economic comparison, it would be more appropriate to present figures in terms of Purchasing Power Parity (PPP). This would allow for a clearer understanding of economic disparities and their impact on healthcare accessibility.
- The section on Knowledge about ASD could be expanded to address adult diagnosis. If many autistic individuals go undiagnosed, is it possible that existing tools are inadequate and based on a narrow, outdated understanding of autism? Given that many autistic individuals do not present developmental delays and may even exhibit advanced cognitive abilities, this should be considered in the discussion.
- In section 4.3, a more in-depth critique of the M-CHAT is warranted. Its limitations should be highlighted more explicitly, particularly concerning its reliability and potential for misclassification.
- There are several instances of extra spaces in the text (lines 122, 191, 248, 281...). A careful proofreading would enhance readability and presentation.
Author Response
Dear Reviewer 2, please find below the document with point-by-point response to your comments.
Best wishes,
Felipe Alckmin-Carvalho

Round 2
Reviewer 1 Report
Comments and Suggestions for Authors
Dear authors,
I commend you for the changes in the manuscript. Please, consider these final suggestions to improve the written quality of the article:
1) use the same terminology to describe the nursing professionals in the article. In the abstract, line 20, the term "nurse" is used. I suggest changing it to "nursing professionals (auxiliary, technician, registered nurse)", as well as in the whole article, when appropriate. Different terminologies can confuse the reader.
2) do not use the term "autism" (lines 55, 89, 93, 114 ...), prefer to use Autism Spectrum Disorder or ASD, which is currently the correct term to refer to this condition
Best Regards
Author Response
Dear Reviewer,
Thank you for your comments.
Please find below the answers:
1) use the same terminology to describe the nursing professionals in the article. In the abstract, line 20, the term "nurse" is used. I suggest changing it to "nursing professionals (auxiliary, technician, registered nurse)", as well as in the whole article, when appropriate. Different terminologies can confuse the reader.
(Answer 1) Thank you for the suggestion. We have modified the expression in the whole manuscript.
2) do not use the term "autism" (lines 55, 89, 93, 114 ...), prefer to use Autism Spectrum Disorder or ASD, which is currently the correct term to refer to this condition
(Answer 2) Thank you for the recommendation. We have modified the expression in the whole manuscript.
All the changes are in blue color, in the manuscript.
Best wishes,
Authors.